# Changes in HMO Concentrations throughout Lactation: Influencing Factors, Health Effects and Opportunities

**DOI:** 10.3390/nu13072272

**Published:** 2021-06-30

**Authors:** Caroline Thum, Clare Rosemary Wall, Gisela Adrienne Weiss, Wendan Wang, Ignatius Man-Yau Szeto, Li Day

**Affiliations:** 1AgResearch Ltd., Te Ohu Rangahau Kai, Palmerston North 4474, New Zealand; li.day@agresearch.co.nz; 2Faculty of Medical and Health Sciences, The University of Auckland, Auckland 1142, New Zealand; c.wall@auckland.ac.nz; 3Yili Innovation Center Europe, 6708 Wageningen, The Netherlands; adrienne.weiss@yili-innovation.com; 4Yili Maternal and Infant Nutrition Institute, Inner Mongolia Yili Industrial Group, Co., Ltd., Fengtai District, Beijing 100071, China; wangwendan@yili.com (W.W.); szeto@yili.com (I.M.-Y.S.)

**Keywords:** breast milk, maternal origin, secretor and Lewis blood type, gut microbiota, 2′-fucosyllactose, lacto-*N*-neotetraose

## Abstract

Human milk oligosaccharides (HMOs) are important functional biomolecules in human breast milk. Understanding the factors influencing differences in HMO composition and changes in their concentration over lactation can help to design feeding strategies that are well-adapted to infant’s needs. This review summarises the total and individual concentration of HMOs from data published from 1999 to 2019. Studies show that the HMO concentrations are highest in colostrum (average 9–22 g/L), followed by slightly lower concentrations in transitional milk (average 8–19 g/L), with a gradual decline in mature milk as lactation progresses, from 6–15 g/L in breast milk collected within one month of birth, to 4–6 g/L after 6 months. Significant differences in HMO composition have been described between countries. Different HMOs were shown to be predominant over the course of lactation, e.g., 3-fucosyllactose increased over lactation, whereas 2′-fucosyllactose decreased. Recent clinical studies on infant formula supplemented with 2′-fucosyllactose in combination with other oligosaccharides showed its limited beneficial effect on infant health.

## 1. Introduction

The concentration of human milk oligosaccharides (HMOs) in milk is higher than the amount of protein [1,2], highlighting their importance for the growing infant. HMOs have been recognized as critical functional biomolecules in human milk [3]; thus, research interest into their biological functions has increased dramatically in the last 30 years.

HMOs are soluble complex sugars containing a combination of different monosaccharides. All HMOs carry lactose (Galβ1–4Glc) at the reducing end linked to a different combination of monosaccharides; D-galactose (Gal), N-acetyl-D-glucosamine (GlcNAc), L-fucose (Fuc) and the sialic acid N-acetylneuraminic acid (Neu5Ac) (Table 1). The chemical structures of 162 HMOs have been characterized to date and are listed in a recent review article by Urashima et al., 2018 [4].

While HMO chemical structure follows a basic blueprint, it has been reported that every woman synthesizes and secretes a distinct HMO profile and has a different individual HMO concentration profile that may be affected by maternal genetics (secretor status) [5], physiology [6], diet [7] and country of origin [8]. Although the effects of this distinct HMO composition on infant health outcomes have yet to be fully understood, a potential link between specific HMOs, milk microbiota and infant’s gut microbiota composition has been described [9,10,11]. These HMO-microbe links have received a considerable amount of research interest in the past decade.

Recently, HMOs were produced by chemical synthesis from raw material or by genetically engineered bacteria [12,13]. The safety of these biotechnologically produced HMOs has been confirmed in many in vitro and in vivo studies [14,15,16,17]. Clinical studies have also been undertaken, and the safety and potential beneficial effects of HMO dietary supplementation have been reported [18,19,20].

This narrative review summarises the concentrations of total and individual HMOs from studies reported in the last 20 years. The selection of manuscripts to be included in this review was based on the criteria published in Thurl et al., 2017 systematic review (Appendix A). Criteria included (i) absolute quantitation of single structures, (ii) milk samples from individual, healthy mothers and (iii) documentation of lactation days. Studies were excluded when samples were pooled, or lactation time for sampling was not clearly stated. Due to the methodological disparities in breast milk sampling and HMO analytical methods employed, a meta-analysis of the collected data is not warranted. Instead, the possible links between maternal genetics, physiology (e.g., health status, environmental factors), country of origin, and the profile of HMO are discussed. The current knowledge on the effects of different HMO profiles on milk and infant gut microbiota and a summary of clinical studies assessing the impact of infant formula supplemented with specific HMOs are also presented.

## 2. Composition throughout Lactation

Every woman secretes a distinct HMO profile containing a particular subset of the 162 HMO structures characterized, and this individual profile remains relatively constant throughout lactation [21]. Among all HMOs structurally characterized, quantitative data are only available for about 30 of them, which represent a significant proportion of the total HMO level (greater than 90%) in breast milk [22].

For the purpose of this review, the chemical names and abbreviations of key HMOs are shown in Table 1. The studies included for integrating the oligosaccharide compositions in human milk are listed in Table 2. The publications were selected based on the 21 studies included in the systematic review conducted by Thurl et al., 2017 [2] and an additional 12 articles reporting data from individual milk samples published between 2016–2019. Table 2 also provides the maternal country of origin, secretor status identified, the number of mothers involved in each study and methods used to quantify HMOs. The reported data from the mothers delivering pre-term (<37 weeks of gestation) were excluded from this review. It is worthy to note that among the 12 HMO composition studies published after 2016, eight of these studies were co-authored and co-funded by commercial companies, demonstrating the increasing research interest in HMO by commercial companies in recent years.

### 2.1. Total HMO Concentration

The total HMO concentration from 16 publications (2007–2019) with documented lactation stages is summarised in Table 3. The total HMO concentrations were either taken from those reported directly in the literature or calculated as the sum of individual HMOs. Studies described in Table 2 that reported total HMO concentrations for fewer than nine HMOs were not included in Table 3.

The data collected from these publications represents studies conducted in different parts of the world, including the USA, Canada, Europe, Asia (China, Japan, Malaysia) and the Pacific region (Samoa). The data show large variations in HMO concentrations in breast milk between individuals within a study and between the studies. All studies show that the HMO concentrations are highest in colostrum (average 9–22 g/L), followed by slightly lower concentrations in transitional milk (between postnatal day 8 and 15, average 8–19 g/L), and a gradual decline in mature milk as lactation progresses, from 6–15 g/L in breast milk collected within one month of birth, to 4–6 g/L in 6 months. The only exception to this is the study conducted by Kunz et al., 2017 in Spain [31], which showed no differences in the total HMO concentrations between colostrum, transitional and mature milk (or between term and preterm milk). Authors suggested that differences in sample preparation and data analysis may explain these discrepancies.

Recently, there has been increased interest in the composition of breast milk in the Chinese population. Several studies have been conducted in collaboration with global and local dairy companies. The data from these studies conducted in the Chinese population varied considerably. The results reported by Elwakiel et al., 2018 [1] were over twice as high as the results reported by Huang et al., 2019 [24] and Ma et al., 2018 [25] at each stage of lactation (Table 3). This may be explained, at least in part, by the different analytical methodologies and the times of breast milk collection used to analyze HMOs. The study conducted by Elwakiel et al., 2018 [1], used capillary electrophoresis-laser-induced fluorescence (CE-LIF), while the study reported by Ma et al., 2018 [25] used HPLC-MRM-MS, and the study by Huang et al., 2019 [24] used the UHPLC-FL method.

The different geographic regions and potentially ethnic diversities in China where the milk samples were collected could also explain the variation in the concentrations of HMOs, e.g., the breast milk samples in the study by Elwakiel et al., 2018 [1] were collected in Hohhot (North of China), in Beijing (Northeast) by Huang et al., 2019 [24] and in Guangzhou (South) by Ma et al., 2018 [25]. However, when breast milk samples were collected from 3 different cities in China (from Northeast to South: Beijing, Suzhou, Guangzhou) and analyzed within the same study by Austin et al., 2016 [32], no difference was observed in the HMO compositions. Given this study did not include the North region of China (e.g., Inner Mogolia/Hohhot), it is still unknown whether there may be any differences in the concentrations and composition of HMOs between different ethnic Chinese groups or populations with different dietary patterns (which have not been reported in the studies). It is important to note that the study by Austin et al., 2016 [32] only reported ten individual HMOs, therefore the total HMOs based on the sum of the ten individual HMOs were lower than any of the results reported by Elwakiel et al. [1], Huang et al. [24] and Ma et al. [25] and cannot be directly compared (Table 3).

There are significant limitations to the compatibility and interpretability of the published studies due to the large differences in milk collection methods, sampling time, and number of HMOs reported and HMO analytical procedures taken by each study. The issues due to the disparities in breast milk sample collection and challenges in establishing accurate and standardized HMO measurements were raised and discussed recently by van Leeuwen, 2019 [53]. Leeuwen, 2019 [53] and others [54,55] have recently reviewed the challenges and pitfalls of HMO analysis, concluding that it is very difficult to compare various studies due to great differences in methods employed to analyze HMOs (sample preparation, HMO separation and detection). All methodologies have a risk of specific losses of HMOs structures, introducing methodological bias. Most techniques have not been extensively assayed for specific HMO losses so a comparative analysis cannot be drawn. It has been consistently suggested the need of a double-blind multi-center study of HMOs analysis to assess methodological bias and the true levels of HMOs in human milk [2,53,54,55].

Moreover, in general, studies only report the results from one region or country (except China and USA), which does not provide a complete representation of maternal ethnicity and/or place of residence effects on HMO profile.

### 2.2. Individual HMO Concentrations

Table 4 summarises the concentrations of the main individual HMOs in colostrum, transitional and mature milk from selected studies covering populations from different countries. These data were taken from individual studies since 2016, the summary data from Thurl et al., 2017 [2] and a number of selected reports between 2007 and 2015 that have measured nine or more HMOs (Table 2). Table 4 contains HMOs reported in at least three publications. The full table, containing all reported HMOs, is shown in the Appendix A. Large standard deviation reported for some HMOs, specially the low abundance ones, can be observed in many studies; demonstrating a substantial variation in the concentration of these HMOs.

As seen for total HMO, the data show large variations in HMO concentrations between individuals within a study and between the studies. The different methods of quantification limit the ability to compare the results from the different publications. However, some general trends can be seen in these data.

2′-FL is the most abundant HMO, accounting for about 20–40% of total HMO in colostrum, except for the Malaysia/Chinese [25], Japanese (except for day 1) [44,45] and Samoan [42] populations in which the average concentration of 2′-FL was slightly lower than LNT or LNFP I (Table 4). The second most abundant HMOs in colostrum are LNDFH I, LNT, LNFP I and 3-FL, each accounting for 10–30% of the total HMOs, followed by a group of the sialylated acidic HMOs 3′-SL, 6′-SL, LST c and DSLNT, and the two neutral HMOs LNnT and LDFT, each accounting for around 2–7% of total HMOs.

Individual HMO concentrations vary during lactation resulting in different HMOs being predominant at a specific stage of lactation. Table 5 provides a summary of the trends regarding changes of individual HMO concentrations throughout lactation. Most HMOs declined as lactation progresses; one exception is 3-FL, which increases throughout lactation. In fact, production of 2′-FL and 3-FL appears to be negatively correlated. This correlation is demonstrated by the collective data from the studies carried out between 1999–2015 [2] and two recent studies [25,32], as illustrated in Figure 1. The results show a strong correlation between 2′-FL and 3-FL concentrations throughout lactation, with R^2^ values from 0.78 to 0.99. Such a strong association indicates a co-regulation between the enzymes responsible for the synthesis of 2′-FL and 3-FL or competition for a limited supply of the same substrate (i.e., guanosine 5′-diphosphate (GDP)-l-fucose).

The expression and activity of the enzymes fucosyltransferases are well known to determine the concentration of fucosylated oligosaccharides in milk [56]. However, given that non-secretors were reported to have increased levels of 3-FL compared to secretors [32,41,57], it may suggest that competition between the fucosyltransferases for substrate also determines the relative levels of the fucosylated oligosaccharides. A limiting effect of the substrate guanosine 5′-diphosphate (GDP)-l-fucose on the total amount of milk fucosylated oligosaccharides has been suggested [32].

The concentrations of LNT and LNnT decrease throughout lactation [2,25,32]; still, levels reported in the literature vary significantly (Table 4). The core structures LNT and LNnT can be elongated via additions of fucosyl- or sialyl-residues forming many other HMOs. Additionally, other core structures could be formed by additions of galactosyl and N-acetylglucosaminyl residues to LNT and LNnT. Of these two core structures, LNT is the predominant. The predominance of type-I structures (those containing the Gal-β-1,3-GlcNAc moiety) over type-II structures (containing the Gal-β-1,4-GlcNac moiety) is exclusive to human milk [58].

Findings on the HMO concentrations over the stages of lactation and clusters based on 2′-FL concentrations suggest that LNT and LNnT are ‘co-regulated’ with the enzyme α1-2-fucosyltransferase (FUT2), with LNT showing a negative and positive correlation with 2′-FL and LNnT, respectively [30]. Although a relatively substantial variation in HMOs between the high and low 2′-FL clusters has been reported, differences in HMO profiles were shown to have no short-term impact on infant growth [30]. Long-term health effects of the different HMO profiles, however, have not been investigated.

At early stages of lactation (<3 months), 6′-SL is the predominant form of sialylated HMO (137–1770 mg/L) (Table 4). As lactation progresses beyond 2–4 months, the concentrations of 6′-SL and 3′-SL become comparable, with the concentration of 3′-SL being higher at 4–8 months [59,60]. The data presented in Table 4 show that LST c is also a dominant sialylated HMO at the very early stage of lactation, i.e., in colostrum (LSTc, 480–1326 mg/L). It rapidly decreases as lactation progresses.

In general, the changes in individual HMO concentrations throughout lactation impact the proportions of fucosylated, non-fucosylated neutral HMOs and sialylated acidic HMOs. Xu et al., 2017 [29] measured HMOs in milk from 45 individual women in the United States collected on postnatal days 10, 26, 71, and 120. They found that the absolute concentrations of total HMOs and of the various types (fucosylated, non-fucosylated neutral HMOs and sialylated acidic HMOs) decreased throughout lactation. The percentage of non-fucosylated neutral and percentage of sialylated HMOs decreased; however, the percentage of fucosylated HMO increased significantly from 60.9% to 77.4% (*p* < 0.05) during the lactation period. Similar results over time were observed by grouping the data on individual HMOs generated by Thurl et al., 2017 [2] (Figure 2A). The percentage of non-fucosylated neutral HMOs, however, remained stable over the lactation period reported (lactation day 5 to 100) (Figure 2B).

## 3. Factors Influencing HMO Profile and Concentration

### 3.1. Secretor and Lewis Blood Group Status

Every lactating woman synthesizes a different set of oligosaccharides from among the 162 HMO structures characterized so far [61]. Some of the variations in HMO composition can be explained by maternal genetics, e.g., secretor status [1]. The secretor status is determined by the expression of certain glycosyltransferases, in particular, the fucosyltransferases, encoded by the secretor (Se) and Lewis (Le) genes, both determine the profile and relative abundance of HMOs [62].

There are four milk groups, determined by the distinct activity of the two enzymes FUT2 and the α1-3/4-fucosyltransferase FUT3 (encoded by the Se and the Le gene, respectively) [5,43,63]. Abundance of α1-2-fucosylated HMOs, especially 2′-FL, is found in the milk of secretors (Se+) while non-secretors’ milk, due to the lack of FUT2 enzyme, does not contain, or contains a minimal amount, of 2′-FL and other α1-2-fucosylated HMOs [5]. A ‘weak secretor’ has been reported in some Asian populations [23,25,30,32] which, due to modifications in the amino acid sequence, produces FUT2 with significantly reduced activity [64]. The HMOs 2′-FL2′-FL and LNFP I, for example, may be present in the milk in lower concentrations than those characteristically found in the milk of secretor mothers [1].

The distribution of secretors in different countries is presented in Table 6. The recent cross-sectional study by McGuire et al., 2017 [6] collected breast milk from a total of 410 women in 11 international populations. It showed that the proportion of women categorized as being secretors varied from 65% in populations in the rural Gambia and rural Ethiopia to 85% and 78%, respectively, in urban populations. Interestingly, the relative amount of secretors in the North American Caucasian populations was lower (66–77%) [6,8,27,29] compared to Hispanic populations living in the USA (95%) [6] or South American (84–100%) populations [6,8]. European countries have also shown higher percentages of secretors (76–100%) [6,8,42] than North American Caucasians. Asian populations, including the Philippines (46%) [8], had the lowest proportion of secretors (46–79%) [1,8,25,32] reported so far.

Maternal secretory status was shown to affect HMO concentrations in different lactation stages. Xu et al., 2017 [29] reported that the total concentration of HMOs in secretors in the USA was to some extent higher (6.3–18%) than that in non-secretors at lactation days 10, 26, 71 and 120. Changes in absolute concentrations of fucosylated, sialylated, and non-fucosylated neutral HMOs are also presented. As expected, fucosylation was 14–39% higher in secretors milk compared to non-secretors, at all postnatal days tested. Sialylation and non-fucosylated neutrals, however, were 25% lower in secretor than non-secretor mothers on lactation day 120 and day 10, respectively (*p* < 0.05) [29].

A similar trend was observed for the 88 mothers from Malawi [29], which consisted of 69 secretors (78%) and 19 non-secretors (22%), at six months postnatal. The total concentrations of HMOs in the milk of the secretors (6.5 ± 1.7 g/L) were significantly higher than those in non-secretors (5.2 ± 2.5 g/L) (*p* < 0.05). The total fucosylated HMOs concentration was higher (4.9 ± 1.2 compared with 3.4 ± 2.3 g/L; *p* < 0.05) and the sialylated and non-fucosylated neutral HMOs were lower in secretors’ milk in both absolute and relative terms.

The differences in Chinese women’s HMO compositions classified as secretor and Lewis positive (sub) groups were reported by Elwakiel et al. [1]. of the total samples, 73% (30) were in the Se+Le+ group (22), while 20% were assigned to the Se−Le+ and 7% to the Se+Le− groups. Higher concentrations of total neutral fucosylated HMO fraction were found in the Se+ groups compared to the Se− group (Figure 3). In this study the ratios of total acidic to total neutral HMO concentrations were also calculated. This showed variation between 13:87–12:88 and 28:72–40:60 over lactation for the Se+Le+ milk-type and Se−Le+ milk-type groups, respectively, indicating that in Se−Le+ mothers, acidic HMOs might be relatively more dominant over time than in Se+Le+ mothers (Figure 3).

The study by Austin et al., 2016 [32] also on Chinese mothers, showed that non-secretor milk tends to have a higher 3-FL concentration than that of secretor milk during the lactation period tested in the study (5 days to 8 months). Given this study was conducted on a larger number of samples (n 446) compared to others (n 20–40), the results provide more robust evidence of such a relationship. The authors suggest that the relative levels of the fucosylated HMOs result from the competition between the enzymes for a limited supply of substrate.

### 3.2. Country of Origin

With the improvement of analytical methods, more data have emerged in the last few years on HMO composition from mothers who live in various parts of the world (Table 3 and Table 4). However, due to the large variations in the data presented (or displayed) from each study and in the sampling procedures used in particular regions or on different subpopulations, it is challenging to compare the data between studies. Such comparative studies will need to be carried out on a large population cohort within a single study and/or with a standardized and validated inter-laboratory methods.

The most extensive single study across countries was reported by Erney et al., 2000 [8]. The authors analyzed neutral oligosaccharides in 549 human milk samples from 435 women residing in 10 countries (Chile, France, Germany, China (Hong Kong), Italy, Mexico, the Philippines, Singapore, Sweden, and the United States). The study found some differences in oligosaccharide profiles between women from different countries and continents. All samples contained HMO structures based on LNT and LNnT; however, none of the fucosyl-oligosaccharides were detected in 100% of the samples. For example, 100% of the samples from Mexico (*n* = 156) contained 2′-FL and LNFP I, whereas only 46% of the Philippines samples contained these two oligosaccharides (*n* = 22). The authors attributed the different HMO profiles between geographical regions to evolutionary-driven genetic differences (secretory status) between different countries’ inhabitants. The authors, however, emphasized that most of these comparisons were weak because of the relatively small sample sizes for each country.

Significant differences in the HMOs (2′-FL, 3-FL, LNFP I, LNFP III, and LNDFH II) between countries were also found in the recent study by Gomez-Gallego et al., 2018 [28], who analyzed the HMOs in 79 women milk from Finland, Spain, South Africa, and China (lactation time unknown), as part of the analysis of human milk metabolites using NMR. The study showed that compared to breast milk samples from Finland, the Chinese samples exhibited significantly higher levels of 3-FL and LNFP III while South Africans showed higher levels of 3-FL. A lower abundance of 2′-FL and LNFP I was observed in Chinese populations [28]. This is in agreement with the low abundance of secretors reported for Asian populations (67–79%) [1,8,25,32] compared to European (78–100%) [6,8,43].

The data we reviewed from the 15 recent studies (Table 4) appears to support the finding by Gomez-Gallego et al., 2018 [28] and likewise the distribution of secretors and non-secretors discussed above. The 2′-FL concentrations were higher in the breast milk samples from women living in EU countries and in the USA (at 2210–4130 mg/L in colostrum [21,26,31,34], 2061–3370 mg/L in transitional [31,34,43,65] and 1753–3020 mg/L in 1 month mature milk [21,34,43,66], respectively) compared to those in samples from women in Asian regions (i.e., China, Malaysia, Japan, Singapore and Samoa) (at 1580–2490 mg/L in colostrum [24,25,44,45], 220–2000 mg/L in transitional [24,25,32,42] and 1371–2170 mg/L in 1 month mature milk [24,25,30,32], respectively). However, there were no clear trends among countries or regions for 3-FL or other HMOs in the data presented in Table 4.

### 3.3. Maternal Physiological Status

In addition to maternal genetics, maternal health and environmental factors may also affect HMO composition. For example, some preliminary data reported by Bode, 2019 [7] suggests that obesity or chronic inflammatory diseases could impact HMO composition [7].

To date, only a few studies have examined the effect of maternal diet on HMO composition. The CHILD cohort study (Azad et al., 2018 [27]) showed that diet quality (Health Eating Index-2010 score) was not correlated with total HMO concentrations, although there are a few dietary components that were associated with individual HMOs. The consumption of whole grains was positively correlated with fucosyllacto-*N*-hexaose while the consumption of total protein and empty calories was negatively correlated with LST b concentration. Additionally, energy intake was positively correlated with LNT and DFLNH concentration. However, the authors highlighted that these associations were relatively weak and perhaps a larger test samples should be required to establish and correlation. A more detailed assessment of nutrient intake during lactation may be required to identify (or exclude) dietary effects on HMO composition. The cross-sectional data also indicate that parity increases overall HMO concentration, but maternal age, delivery method, or infant gender showed no association with HMO composition [27].

The study by McGuire et al., 2017 [6] examined the relationships between HMO and maternal anthropometric and reproductive indexes from 11 international populations. It investigated whether compositional differences were related to environmental variations, in addition to genetics. The study found that maternal age, weight, and body mass index (BMI) were correlated with the concentration of many HMOs. Additionally, populations from similar ethnicity (and likely genetics) but living in different locations showed significant differences in HMO concentrations (e.g., LNnT and DSLNT), suggesting that the environment, specifically maternal nutritional factors play a role in regulating the synthesis of HMOs. The authors conclude that average HMO concentrations and profiles vary geographically. Targeted genomic analyses are needed to determine whether these differences are due at least in part to genetic variation.

The latest study by Samuel et al., 2019 [21] from mothers across 7 European countries (*n* = 290), showed that maternal pre-pregnancy BMI, mode of delivery and parity determined minor but significant differences in HMO concentrations. Their findings suggest that HMO composition is regulated time-dependently by enzyme activity and substrate availability. It was also suggested that maternal physiology may influence glycosylation within the initial period of lactation.

The information on how maternal diet may influence HMO composition has been recently reviewed [67]. Although positive associations between diet, nutrition status and HMO profile has been found, robust data remains scarce. Suitable studies are necessary to explore possible alterations in HMO composition due to maternal dietary, caloric, and nutrient intake. A careful examination of sociocultural, behavioural, and environmental factors also needs to be considered to determine their roles in this regard. The role of diet, exercise, and other lifestyle factors impacting HMO composition in breast milk is currently under investigation by research groups [7].

## 4. Health Effects of HMO

Due to their indigestible characteristics and structural similarity with mucosal glycans, HMOs are expected to affect numerous glycan-mediated processes such as colonization of the GI tract by early-life microbiota, development of the immune system and the infectivity of pathogens [68,69]. Based on clinical, in vivo, and in vitro studies, HMOs act in a structure-function specific way to assist:The establishment of a mucous-adapted microbiome, by acting as a preferred substrate for the growth of selected “good” bacteria [70,71,72,73,74,75]Directly modulating immune responses by acting either locally on cells of the mucosa-associated lymphoid tissues or systemically to inhibit the expression of inflammatory genes [76,77,78]Resistance to pathogens, by acting as decoy molecules that are bound by pathogenic bacteria, preventing the bacteria from binding to the surface of the host cells [75,79,80];

Two articles published in the recent Nestle Nutrition Institute Workshop Series (2019) provided comprehensive reviews on the most recent research and development related to HMOs [7,68]. Two other earlier reviews studies, one by Kobata, 2000 [61] and other by Bode, 2012 [3] provide an excellent overview of the history of HMO research and knowledge gained at different periods, HMO structural diversity, what is known about HMO biosynthesis in the mother’s mammary gland and the postulated beneficial effects of HMO for the breast-fed infant. In this review we will focus on the effects of HMO profile and concentration on the infant gut and maternal milk microbiota.

### 4.1. HMO Profile and Microbiota in Infant’s Gut and Mother’s Milk

To date, significant evidence has been presented to demonstrate an association of HMO composition with the gut microbiota in infants. A healthy infant gut microbiome is often dominated by *Bifidobacterium* species. These bacteria can represent up to 90% of the total infant microbiome. The proliferation of *Bifidobacterium* in a breast-fed infant can be explained partly by the high amounts of HMO in breast milk [81]. Infant diet is one of the critical factors that shape the early-life microbiota. It has been established that breast-fed and formula-fed infants differ in microbial composition and microbial diversity, including significant differences in bifidobacterial populations [82], which has also been linked to differential health outcomes, e.g., induction of allergies and asthma [83].

Several studies have reported positive correlations between total HMO concentration and milk bacteria, including *Bifidobacterium* spp. and *Staphylococcus*. Recently Moossavi et al. (2019) conducted an integrative analysis of milk microbiota with HMOs and fatty acids using a sub-sample of 393 mothers in the Canada CHILD birth cohort [66]. Milk samples were collected at 3–4 months postpartum and microbiota in milk were analyzed using 16S rRNA gene V4 sequencing. The study found that oligosaccharides (FLNH, LNH, LNFP I) were associated with milk microbiota diversity, while two sialylated HMOs—3′-SL and DSLNT, were associated with overall microbiota composition. Notably, *Bifidobacterium* prevalence was associated with lower abundances of DSLNH.

The HMO profile also has a role in shaping the infant’s gut microbiota. It has been shown that secretor status correlates with a higher abundance of *Bifidobacterium* species in the gut microbiome of infants receiving this breast milk [84]. In contrast, infants fed with non-secretor milk showed a delay in the colonization by these beneficial microorganisms and more *Clostridium* and *Enterobacteria* in their faeces [84].

As part of the MING study by Austin et al., 2016 [32], microbiota profiles in breast milk of Chinese lactating mothers at different stages of lactation were examined and published by Sakwinska et al., 2016 [65]. Microbiota profiling was based on the sequencing of fragments of 16S rRNA gene and specific qPCR for bifidobacteria, lactobacilli and total bacteria to study microbiota of the entire breast milk collected using a standard protocol without aseptic cleansing (*n* = 60), and the microbiota of the milk collected aseptically (*n* = 30). The study found that the microbiota of breast milk was dominated by streptococci and staphylococci for both collection protocols. There were higher bacterial counts in the milk collected using the standard protocol compared to the milk collected aseptically. Bifidobacteria and lactobacilli were present in few samples with low abundance. The study found no effect of the stage of lactation or the delivery mode on microbiota composition. We believe this is the first, and only reported study of human milk microbiota from the Chinese population.

Recently Moossavi et al., 2019 [66] conducted an integrative analysis of milk microbiota with HMOs and fatty acids using a sub-sample of 393 mothers in the Canadian CHILD birth cohort [66]. Milk samples were collected at 3–4 months postpartum and milk microbiota was analyzed using 16S rRNA gene V4 sequencing. Oligosaccharides and fatty acids were analyzed. The study found that oligosaccharides (LNDFH, LNH, LNFP I) were associated with microbiota α diversity (the observed richness (number of taxa) or evenness (the relative abundances of those taxa) of an average sample), while two sialylated HMOs—3′-SL and DSLNT, were associated with overall microbiota composition. Notably, *Bifidobacterium* prevalence was associated with lower abundance of DSLNH. Additionally, among non-secretor mothers, *Staphylococcus* was positively correlated with sialylated HMOs. Overall, the relationships between HMOs and the microbiota in milk were not as strong as the correlation between the overall milk fatty acid profile and some individual long chain fatty acids (22:6 n-3, 22:5 n-3, 20:5 n-3, 17:0, 18:0) with milk microbiota composition. While there appears to be only small collective effect of HMOs on the milk microbiota, the authors suggest that individual HMOs might promote or inhibit growth of specific milk bacteria, potentially providing a selection mechanism for vertical mother-offspring transmission of microbiota.

Overall, these interesting results highlight the need to examine HMOs and milk microbiota in larger studies using standardized protocols for the collection and analyses of milk, while accounting for secretor status, mother’s diet, physiological status and other potential confounding factors.

### 4.2. Supplementation of Infant Formula with HMOs

Milk oligosaccharides in human milk are 100–1000 times higher than those found in ruminant milk (e.g., cows, goats, and sheep). Not only the concentration but also the profile of oligosaccharides in human milk is unique and more complex compared to farm animals [85]. Therefore, infant formula products based on cow’s milk lack the oligosaccharide composition and concentrations naturally present in human breast milk.

Although the role of oligosaccharides as the bifidogenic factor in human milk was identified in the 1930s [70] and the main HMO structures were reported in 1954 [86], it was only in the late 2010s that researchers were able to produce the first oligosaccharide structurally identical to those in human milk at a large scale [12,13]. Such progress has made available many individual HMOs, namely 2′-FL [87], LNT [88] and LNnT [89], 3-FL [90], 3′-SL, and 6′-SL [91].

Today, HMOs are classified as novel foods by many food regulation authorities requiring a mandatory food safety assessment. So far, eight HMOs (2′-FL [92], 3-FL, LNnT, DFL, LNT [93], 6′-SL and 3′-SL [94]) have being positively assessed and approved by the European Food Safety Authority (EFSA). The combination of 2′-FL and LNnT or DFL [92,95] was also regarded as safe for infants up to one year of age when added to infant and follow-on formulae, based on the scientific and technical information provided. EFSA has approved the use of HMO in a range of foodstuffs, such as some dairy products, cereal bars, foods for special medical purposes, or flavoured drinks.

In the USA, HMOs intended for use in foods other than dietary supplements can be approved by a panel of qualified scientists, with or without FDA advice. This happens under the same regimen as all other food ingredients–that is, they may be introduced, at the discretion of the manufacturer, as food additives or as “Generally Regarded as Safe” (GRAS) substances. So far, GRAS status has been decreed for 2′-FL (GRAS note 650), 3-FL (GRAS note 925), LNnT (GRAS note 919), DFL, LNT (GRAS note 923), 6′-SL (GRAS note 922), 3′-SL (GRAS note 921) and for the combination of 2′-FL and DFL (GRAS note 815).

Reverri et al., 2018 [18] recently published a review on the clinical studies using infant formula containing the HMO 2′-FL. The article provided a summary of the results of these clinical trials and additional unpublished results on a clinical feeding study of a partially hydrolyzed whey-based formula. Two placebo-controlled, blinded, randomized, clinical intervention studies were conducted in 28 sites across the USA to show the growth safety (weight, length, and head circumference) and tolerance of 2′-FL combined with either galacto-oligosaccharides (GOS) or fructooligosaccharides (FOS) in infant formula [19,20]. Infants fed with infant formulas supplemented with 2′-FL combined with GOS (0.2 g/L 2′-FL plus 2.2 g/L GOS or 1 g/L 2′-FL plus 1.4 g/L GOS) or FOS (2 g/L FOS plus 0.2 g/L 2′FL) showed similar growth as breastfed infants up to 4 months of age (*n* = 314). These studies [19,20] were the first publications showing that growth of infants consuming a formula containing 2′-FL was similar to that of breastfed infants.

The effects of feeding formulas supplemented with 2′-FL on immune function biomarkers were investigated in a subgroup of the above study population [96]. The results showed decreased concentrations of plasma inflammatory cytokines and Tumour Necrosis Factor alpha (TNF-α) in infants fed with the supplemented infant formula compared to control, resembling that of breastfed infants, at both doses tested.

Another randomized controlled infant trial showed that whey-based infant formula supplemented with 2 HMOs, 2′-FL and LNnT (*n* = 88), allowed for age-appropriate growth of infants and was well tolerated when compared to the same infant formula without HMO [97]. Infants receiving formula supplemented with 1 g/L 2′-FL and 0.5 g/L LNnT had improved sleep quality and softer stools at two months of age, and caesarean section infants had a lower occurrence of colic at four months of age. At 4, 6 and 12 months of age infants receiving the supplemented formula had fewer parental reports of bronchitis and overall decreased episodes of lower respiratory tract infections and use of antibiotics from 6 months of age compared to infants fed with the formula containing no HMOs. Protective effects were still observed after the six months of the intervention period [97]. The study showed an correlation between feeding the 2-HMO supplemented infant formula and lower reported respiratory tract illnesses and medication use (especially antibiotics and antipyretics) during the first year of life. These findings warrant confirmation in further studies.

In the same trial, the infants gut microbiota was also examined. Overall, infants fed the formula with 2′-FL and LNnT developed a gut microbiota closer to the microbiota observed in breastfed infants [68,98]. The supplementation of infant formula with these two HMOs promoted the growth of *Bifidobacterium* and decreased potentially pathogenic bacteria *Escherichia* and *Peptostreptococcaceae* at three months of age. The reduction in antibiotic use by the infant consuming HMO-supplemented formula may also be related to gut microbiota profile observed. At three months, the microbiota profile in the infants fed with 2-HMO supplemented infant formula shifted away from those who were fed with the control formula without 2-HMO and towards those who were breast-fed.

Although there have been only a few clinical studies, these results generally point towards a trend of potential health and wellbeing outcomes of HMO-supplemented formula-fed infants, which are similar to those of breast-fed infants. More prospective and randomized trials are needed to evaluate the health benefits and provide validated evidence of supplementing infant formula with HMOs.

## 5. Conclusions

Considerable variations in HMO concentrations throughout lactation and between individual mothers within and among studies were observed. The variations observed between individuals in the same study are likely due to maternal factors such as Secretor and Lewis blood group status, which are not always taken into account or discussed in the reports. Other maternal factors such as country of origin, BMI and parity may also play a role in HMO profiles, but their influence still needs to be demonstrated in extensive cohort studies. The considerable variation in concentrations (individual and total) of HMOs reported between studies reveals the importance of the need to standardize the milk collection method, processing and analysis of HMOs between research laboratories.

In general, all studies agree that the HMO concentrations are highest in colostrum, followed by slightly lower concentrations in transitional milk, and a gradual decline in mature milk as lactation progresses. Individual HMO concentrations change during lactation with different HMOs being predominant in each stage. 2′-FL is the most abundant HMO, accounting for about 20–40% of total HMO in concentration in colostrum whereas 6′-SL is the dominant form of sialylated HMO at the early stages of lactation followed by 3′-SL in late lactation. Most HMOs declined as lactation progressed, except 3-FL, which was negatively correlated with 2′-FL.

Four milk groups can be assigned based on the Secretor and Lewis blood group system (Se+Le+, Se+Le−, Se−Le− and Se−Le+). The milk of a Secretor is characterized by an abundance of α1-2-fucosylated HMOs, especially 2′-FL. Conversely, the milk of a non-secretor does not contain 2′-FL or other α1-2-fucosylated HMOs, or they are only present in minimal amounts (weak secretor found in some Asian populations). Furthermore, core structures, such as LNT, and acidic HMOs, over time, might be relatively more dominant in non-secretor’s milk.

The distribution of secretors among a population differs depending on maternal origin. Secretors from EU countries and South American were all above 80%, higher than those from Asia, USA (non-Hispanic) and rural Africa. Secretor status correlates with a higher abundance of *Bifidobacterium* species in the infants gut microbiome. Infants fed with non-secretor milk showed a delay in the gut colonization by these beneficial microorganisms. The differences in the secretor status of mothers may lead to differences in colonization of the infant gut by the microbiota and may also be linked with mothers’ milk microbiota composition.

So far, supplementation of infant formula with 2′-FL, alone or in combination with LNnT or other oligosaccharides (e.g., FOS or GOS) was shown to be well-tolerated. While the beneficial effects of HMO addition to infant formula still need to be demonstrated in more randomized trials, recent data show potential beneficial effects on the infant’s immune modulation and gut microbiota colonization.

## Figures and Tables

**Figure 1 nutrients-13-02272-f001:**
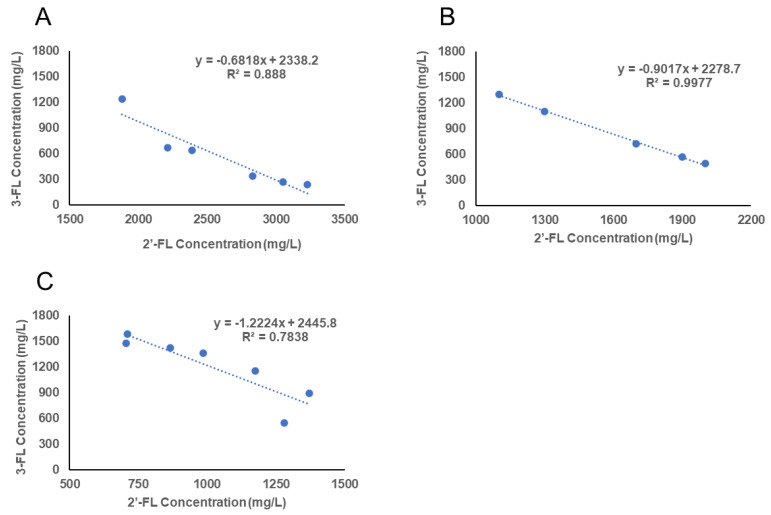
Correlation between 3-fucosyllactose (3-FL) concentration and 2′-fucosyllactose (2′-FL) concentration throughout lactation. (**A**) Collective data on the studies carried out between 1999–2015 from Thurl et al., 2017 [2]). (**B**) Data reported by Austin et al., 2016 [32]; (**C**) by Ma et al., 2018 [25]).

**Figure 2 nutrients-13-02272-f002:**
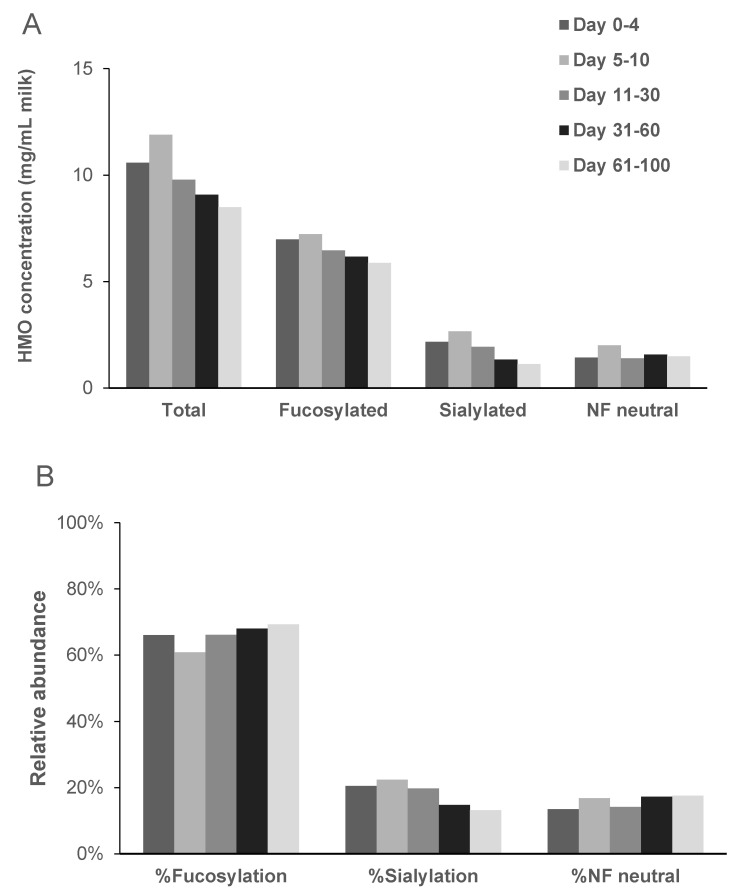
Changes in absolute (**A**) and relative (**B**) concentrations of human milk oligosaccharides (HMOs) from secretor mothers during the course of lactation. (**A**) Total, fucosylated, sialylated, non-fucosylated neutral HMO concentrations decreased over time. (**B**) Percentage of fucosylation increased while percentage of sialylation decreased and non-fucosylated neutrals remained stable over time. NF, non-fucosylated neutral oligosaccharides; The data were compiled by Thurl et al., (2017) [2].

**Figure 3 nutrients-13-02272-f003:**
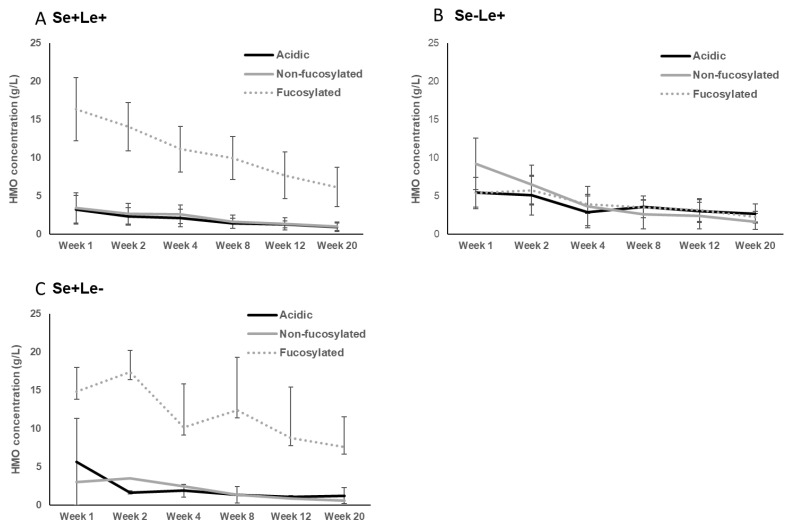
Concentrations of oligosaccharides in Chinese human milk over a 20-wk lactation period for (**A**) Se+Le+ milk-type group *n* = 22, (**B**) Se−Le+ milk-type group *n* = 6, and (**C**) Se+Le− milk-type group *n* = 2. (Adapted from Elwakiel et al., 2018 [1]).

**Table 1 nutrients-13-02272-t001:** List of abbreviations of the most common HMO compounds with their chemical names and structures described in the literature and presented in this review.

Abbreviation	Name	Structure (Monomers and Linkages)
** 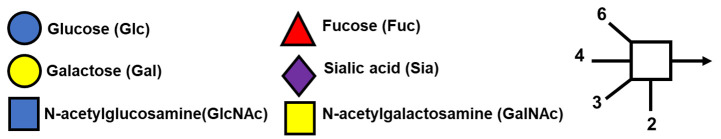 **
Neutral fucosylated
2′-FL	2′-fucosyllactose	Fucα1,2Galβ1,4Glc	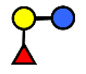
3-FL	3-fucosyllactose	Galβ1,4(Fucα1,3)Glc	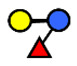
DFLac/LDFT	Lactodifucotetraose	Fucα1,2Galβ1,4(Fucα1,3)Glc	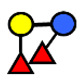
DFLNT	Difucosyllacto-*N*-tetrose	Fuc(α1-4)[Gal(β1-3)]GlcNAc(β1-3)Gal(β1-4)[Fuc(α1-3)]Glc	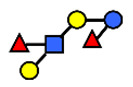
DFLNH I	Difucosyllacto-*N*-hexaose I	Galβ1,4(Fucα1,3)GlcNAcβ1,6(Fucα1,2Galβ1,3GlcNAcβ1,3)Galβ1,4Glc	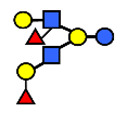
DFLNH II	Difucosyllacto-*N*-hexaose II	Galβ1,4(Fucα1,3)GlcNAcβ1,6(Galβ1,3(Fucα1,4)GlcNAcβ1,3) Gal β1,4Glc	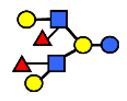
LNDFH I	Lacto-*N*-difucohexaose I	Fucα1,2Galβ1,3(Fucα1,4)GlcNAcβ1,3Galβ1,4Glc	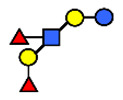
LNDFH II	Lacto-*N*-difucohexaose II	Galβ1,3(Fucα1,4)GlcNAcβ1,3Galβ1,4(Fucα1,3)Glc	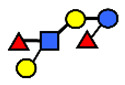
LNnDFH	Lacto-*N*-neodifucohexaose	Galβ1,4(Fucα1,3)GlcNAcβ1,3Galβ1,4(Fucα1,3)Glc	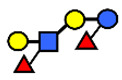
LNFP I	Lacto-*N*-fucopentaose I	Fucα1,2Galβ1,3GlcNAcβ1,3Galβ1,4Glc	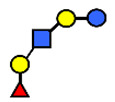
LNFP II	Lacto-*N*-fucopentaose II	Galβ1,3(Fucα1,4)GlcNAcβ1,3Galβ1,4Glc	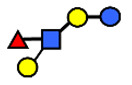
LNFP III/LNnFP II	Lacto-*N*-fucopentaose III Lacto-*N*-fuconeopentaose II	Galβ1,4(Fucα1,3)GlcNAcβ1,3Galβ1,4Glc	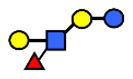
LNFP V	Lacto-*N*-fuconeopentaose V	Galβ1,3GlcNAcβ1,3Galβ1,4(Fucα1,3)Glc	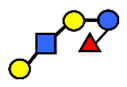
LNnFP I	Lacto-*N*-neofucopentaose I	(Fucα1,2)Galβ1,4GlcNAcβ1,3Galβ1,4Glc	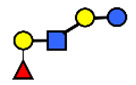
LNnFP V	Lacto-*N*-neofucopentaose V	Galβ1,4GlcNAcβ1,3Galβ1,4(Fucα1,3)Glc	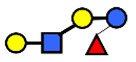
Neutral non-fucosylated
3′-GOS/3′-GL	3′-galactosyllactose	Galβ1,3Galβ1,4Glc	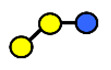
6′-GOS/6′-GL	6′-galactosyllactose	Galβ1,6Galβ1,4Glc	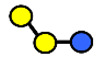
LNT	Lacto-*N*-tetraose	Galβ1,3GlcNAcβ1,3Galβ1,4Glc	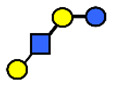
LNnT	Lacto-*N*-neotetraose	Galβ1,4GlcNAcβ1,3Galβ1,4Glc	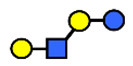
LNH	Lacto-*N*-hexaose	Galβ1,3GlcNAcβ1,3(Galβ1,4GlcNAcβ1,6)Galβ1,4Glc	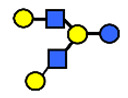
LNnH	Lacto-*N*-neohexaose	Galβ1,4GlcNAcβ1,3(Galβ1,4GlcNAcβ1,6)Galβ1,4Glc	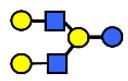
MFLNH I/2-FLNH	Monofucosyllacto-*N*-hexaose I	Fucα1,2Galβ1,3GlcNAcβ1,3(Galβ1,4GlcNAcβ1,6)Galβ1,4Glc	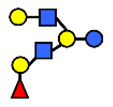
MFLNH III/3-FLNH	Monofucosyllacto-*N*-hexaose III	Galβ1,3GlcNAcβ1,3(Galβ1,4(Fucα1,3)GlcNAcβ1,6)Galβ1,4Glc	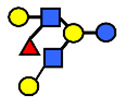
Acidic	
3′SL	α2,3-Sialyllactose	NeuAcα2,3Galβ1,4Glc	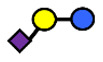
6′SL	α2,6-Sialyllactose	NeuAcα2,6Galβ1,4Glc	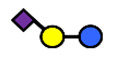
3′S3FL	3′-sialyl-3-fucosyllactose	NeuAcα2,3Galβ1,4(Fucα1,3)Glc	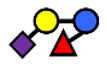
6′SLN	6′-sialyl lactosamine	NeuAcα2,6Galβ1,4GlcNAc	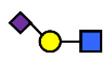
LST a	α2,3-Sialyllacto-*N*-tetraose a	NeuAcα2,3Galβ1,3GlcNAcβ1,3Galβ1,4Glc	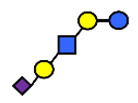
LST b	α2,6-Sialyllacto-*N*-tetraose b	Galβ1,3(NeuAcα2,6)GlcNAcβ1,3Galβ1,4Glc	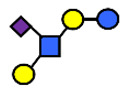
LST c	α2,6-Sialyllacto-*N*-tetraose c	NeuAcα2,6Galβ1,4GlcNAcβ1,3Galβ1,4Glc	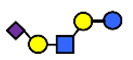
DSLNT	Disialyllacto-*N*-tetraose	NeuAcα2,3Galβ1,3(NeuAcα2,6)GlcNAcβ1,3Galβ1,4Glc	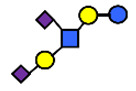
S-LNnH II	α2,6-Sialyllacto-*N*-neohexaose	NeuAcα2,6Galβ1,4GlcNAcβ1,3(Galβ1,4GlcNAcβ1,6)Galβ1,4Glc	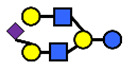
A-Tetra	A-tetrasaccharide	GalNAcα1,3(Fucα1,2)Galβ1,4Glc	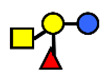
SLNFP I	Sialyllacto-*N*-fucopentaose I	Fucα1,2Galβ1,3(NeuAcα2,6)GlcNAcβ1,3Galβ1,4Glc	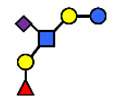
SLNFP II	Sialyllacto-*N*-fucopentaose II	NeuAcα2,3Galβ1,3(Fucα1,4)GlcNAcβ1,3Galβ1,4Glc	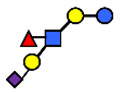

N, a nitrogen-containing disaccharide.

**Table 2 nutrients-13-02272-t002:** Studies on human milk oligosaccharides composition examined in this review.

Reference		Mothers			Quantification
Year of Publication		Country of Origin	Secretor Status	Number of Mothers	Methodology
Studies Published (Post-2015)					
2019	[21]	Europe (Spain, France, Italy, Norway, Portugal, Romania and Sweden)	+/-	9, 83, 13, 10, 95, 40, 40, respectively	UHPLC-FL
2019	[23]	UAE	+/-	40	HPLC-MRM-MS
2019	[24]	China (Beijing)	nr	33	UHPLC-FL
2018	[25]	China (Guangzhou), Malaysia	nr	20 26	HPLC-MRM-MS
2018	[26]	USA	nr	10	HPAEC-PAD
2018	[1]	China (Hohhot) The Netherlands	+/-	30 28	CE-LIF
2018	[27]	Canada	+/-	427	HPLC-FL
2018	[28]	China, South Africa, Finland, and Spain	nr	20, 19, 20, 20, respectively	NMR
2017	[29]	Malawi USA	+/-	88 45	UHPLC/QqQ-MS
2017	[30]	Singapore	nr	50	HPAEC-PAD
2017	[31]	Spain (Valencia)	+/-	14	HPAEC-PAD
2016	[32]	China (Beijing, Suzhou, Guangzhou)	nr	446	UHPLC-FL
Studies reviewed by Thurl et al. [2] ^1^			nr		
2015	[33]	The Netherlands	+/-	7/5	CE-FL
2015	[34]	USA	nr	15/13	NMR
2014	[35]	USA	+/-	13/4	LC-MS
2014	[36]	USA	+/-	10/10	LC-MS
2014	[37]	South Africa	+/-	20/21	HPLC-UV
2013	[38]	USA	+	4	LC-MS
2013	[39]	USA	+/-	40/12	NMR
2011	[40]	Italy	+/-	16/23	HPAEC-PAD
2011	[41]	Italy	+/-	42/21	HPAEC-PAD
2010	[42]	Samoa	nr	16	HPLC-UV
2010	[43]	Germany	+	21	HPAEC-PAD
2008	[44]	Japan	+	12	HPLC-UV
2007	[45]	Japan	nr	20	HPLC-UV
2007	[46]	USA	nr	8	CE-UV
2003	[47]	Spain	nr	12	HPLC-UV
2003	[48]	Japan	nr	20	HPLC-UV
2001	[49]	Mexico	+	11	HPLC-UV
2000	[8]	Various world regions	+	197	HPAEC-PAD
1999	[50]	Italy	+	18	HPAEC-PAD
1999	[51]	Germany	+/-	2/2	HPAEC-PAD
1999	[52]	USA	+	12	HPAEC-PAD

^1^ The systematic review by Thurl et al., (2017) [2] compiled data from 21 articles published between 1999–2015 of HMO concentrations (mean values). The single values at different gestational age and stages of lactation from healthy mothers with identified secretor status were also presented. UHPLC-FL; Ultra-high-performance liquid chromatography with fluorescence detection. HPLC-MRM-MS; High Performance Liquid Chromatography-Multiple Reaction Monitoring-Mass Spectrometry. HPAEC-PAD, High-performance anion-exchange chromatography with pulsed amperometric detection. CE-LIF, Capillary electrophoresis with the laser-induced fluorescence detection. UHPLC/QqQ-MS; Ultra-high-performance liquid chromatography coupled with triple quadrupole mass spectrometry. CE-FL; Capillary electrophoresis coupled with fluorescence detection. NMR; Nuclear magnetic resonance. LC-MS; Liquid chromatography-mass spectrometry. HPLC-UV; High-Performance Liquid Chromatography-with Ultraviolet detector. CE-UV; Capillary electrophoresis with UV detection. Secretor status, (+) secretor; (-) non-secretor; (nr) not reported.

**Table 3 nutrients-13-02272-t003:** Total HMO concentrations (g/L) in human colostrum, transitional, and mature milk from mothers with either positive or unknown secretor status. The concentrations of the HMOs were shown as mean ± standard deviation.

Reference	# HMOs/Monosaccharide Compositions	Mothers	Colostrum	Transition	Mature					
	Measured	Country of Origin	(Day 1–7)	(Day 8–15)	1 Month	2 Months	3 Months	4 Months	5 Months	6 Months
Samuel, 2019 [21]	20	Europe (Spain, France, Italy, Norway, Portugal, Romania and Sweden)	12.5 ± 7.2	11.0 ± 6.0	9.6 ± 5.1	7.6 ± 4.1	6.9 ± 3.7	6.3 ± 3.4		
McJarrow, 2019 [23]	12	UAE	nr	8.2 ± 2.4	nr	nr	nr	nr	nr	3.9 ± 1.4
Huang, 2019 [24]	15	China (Beijing)	9.6 ± 6.1	8.4 ± 5.0	6.6 ± 4.4	nr	nr	nr	nr	nr
Austin, 2016 [32] ^1^	10	China (Beijing, Suzhou, Guangzhou)	5.0 ± 3.6~4.2 ± 2.7		3.5 ± 2.3		3.1 ± 2.0		3.0 ± 2.0	
Ma, 2018 [25]	12	China (Guangzhou)	nr	8.5 ± 4.4	6.1 ± 3.6	5.2 ± 3.1	5.4 ± 3.7	5.2 ± 3.2	nr	4.7 ± 2.9
Ma, 2018 [25]	12	Malaysia	12.5 ± 8.7	nr	nr	6.1 ± 3.3	nr	nr	nr	5.1 ± 3.2
Nijman, 2018 [26]	9	USA	9.1 ± 0.2	nr	nr	6.4 ± 0.29	nr	nr	nr	nr
Elwakiel, 2018 [1]	14	China (Hohhot)	22.4 ± 4.6	18.9 ± 3.9	14.6 ± 4.3	12.4 ± 3.8	10.0 ± 3.7	nr	7.8 ± 3.0	nr
Elwakiel, 2018 [1]	14	The Netherlands	nr	nr	14.7 ± 5.4	nr	nr	nr	nr	nr
Azad, 2018 [27]	19	Canada (Asian and 73% Caucasian)	nr	nr	nr	nr	nr	10.3 ± 6.7	nr	nr
Xu, 2017 [29]	32	Malawi	nr	nr	nr	nr	nr	nr	nr	6.2 ± 2.0
Xu, 2017 [29]	32	USA	nr	19.6 ± 2.9	16.3 ± 2.7	nr	10.4 ± 1.4	8.6 ± 1.3	nr	nr
Kunz, 2017 [31]	16	Spain (Valencia)	7.5 ± 4.1	9.1 ± 3.0	8.2 ± 2.8	nr	nr	nr	nr	nr
Thurl, 2017 [2] ^2^	33	Various	16.3 ± 12.2	17.4 ± 12.2	15.1 ± 9.3	15.1 ± 12.1	nr	13.4 ± 11.5	nr	nr
Spevacek, 2015 [34]	15	USA	13.0 ± 3.9	10.7 ± 2.1	9.2 ± 2.0	nr	nr	nr	nr	nr
Leo, 2010 [42] ^3^	17	Samoa	nr	15.6 ± 8.1	10.9 ± 9.7			nr		
Thurl, 2010 [43] ^3,4^	20	Germany	11.7	11.9	10.7	8.4	8.0	nr	nr	nr
Asakuma, 2007 and 2008 [44,45] ^2^	9, 10	Japan	10.2 ± 5.5	nr	nr	nr	nr	nr	nr	nr

^1^ Transitional (5–30 days), mature milk (4–8 months). ^2^ Data compiled by Thurl et al., 2017 [2] from 21 studies, before 2016. ^3^ Studies compiled by Thurl et al., 2017 [2] but described here as representative of Samoan, German and Japanese populations. ^4^ Data from pooled milk between 1–5 months; HMOs, human milk oligosaccharides; #, number of HMOs measured; nr, not reported.

**Table nutrients-13-02272-t004a:** 

(a) Colostrum	Day 1	Day 2		Day 3			Day 0–4		Day 0–5	Day 1–7	
Reference Year	[44,45] 2007 and 2008	[44,45] 2007 and 2008	[25] 2018	[21] 2019	[44,45] 2007 and 2008	[43] 2010	[26] 2018	[2] 2017	[34] 2015	[31] 2017	[24] 2019
Country	Japan	Japan	Malaysia /Chinese	7 European countries	Japan	Germany	USA	Various	USA	Spain	China
2′-FL	2490 ± 1220	2010 ± 1070	2249 ± 1764	3691 ± 1941	1580 ± 730	4130	3750 ± 100	3230 ± 610	2652 ± 2222	2210 (0–4690)	1705 ± 1101
3-FL	260 ± 140	280 ± 260	429 ± 419	422 ± 453	200 ± 130	240	nr	240 ± 100	444 ± 513	750 (0–1190)	353 ± 305
LNDFH I	1270 ± 540	1870 ± 1550	nr	1232 ± 519	1410 ± 660	1120	2100 ± 60	860 ± 240	nr		798 ± 570
LNDFH II	17 ± 18	20 ± 25	nr	nr	19 ± 28	100	nr	80 ± 90	nr	60 (10–250)	nr
LDFT	420 ± 420	280 ± 300	nr	607 ± 558	190 ± 140	490	360 ± 10	520 ± 270	159 ± 152	180 (0–400)	nr
LNT	890 ± 430	1440 ± 700	2393 ± 2192	912 ± 802	1450 ± 730	nr	480 ± 0	660 ± 410	1054 ± 984	840 (620–1600)	1123 ± 776
LNnT	400 ± 90	540 ± 140	1420 ± 1032	307 ± 132	420 ± 150	nr	nr	770 ± 830	255± 113	310 (140–450)	616 ± 239
LNFP I	1470 ± 1010	2080 ± 1670	3563 ± 1920 ^2^	1928 ± 903	1670 ± 1030	2000	1810 ± 30	1570 ± 300	1409 ± 1153	950 (0–1300)	1509 ± 1032
LNFP II	380 ± 2401 ^1^	450 ± 260 ^1^		422 ± 518	420 ± 330	140	nr	220 ± 190	401 ± 461	150 (0–1510)	365 ± 409
LNFP III				445 ± 166		340	nr	260 ± 290	359 ± 188	380 (260–560)	nr
LNFP V	nr	nr		108 ±103	nr	nr	nr	3 ± 1223	nr	nr	60 ± 75
3′-SL	362± 103	269 ± 70	222 ± 83	254 ± 90	259 ± 80	350	110 ± 10	220 ± 140	228 ± 63	230 (160–330)	228 ± 78
6′-SL	342 ± 120	371 ± 115	651 ± 411	543 ± 168	397 ± 86	1310	340 ± 30	760 ± 580	520 ± 152	680 (500–800)	1175 ± 495
LST a	107 ± 85	155 ± 118	160 ± 111	nr	162 ± 111	60	nr	120 ± 80	nr	150 (110–240)	nr
LST b	68 ± 22	64 ± 25		79 ± 40	61 ± 27	50	nr	110 ± 160	nr	40 (20–50)	nr
LST c	659 ± 297	707 ± 261	1326 ± 641	497 ± 218	693 ± 243	480	nr	480 ± 150	nr	380 (290–440)	743 ± 255
DSLNT	480 ± 126	447± 110	nr	405 ± 178	459 ± 151	290	nr	550 ± 510	nr	380 (240–540)	804 ± 721
SLNFP I	76 ± 55	80 ± 37	nr	nr	78 ± 65	nr	nr	nr	nr	nr	nr
3′S3FL	148 ± 44	156 ± 64	23.4 ± 32.2	nr	165 ± 44	nr	nr	nr	nr	nr	nr

**Table nutrients-13-02272-t004b:** 

(b) Transitional	Day 5–15									
Reference Year	[43] 2010	[43] 2010	[42] 2010	[34] 2015	[32] 2016	[2] 2017	[31] 2017	[25] 2018	[23] 2019	[24] 2019
Country	Germany (8 day)	Germany (15 day)	Samoa (5–10 day)	USA (14 day)	China (5–11 day)	Various (5–10 day)	Spain (8–15 day)	China (14 day)	UAE (5–15 day)	China (8–15 day)
2′-FL	3370	3040	220 ± 370	2061 ± 1416	2000 ± 1400	3050 ± 710	2340 (0–3860)	1281 ± 1050	2021 ± 1776	1507 ± 898
3-FL	260	380	1670 ± 820	933 ± 567	490 ± 600	270 ± 120	950 (0–1430)	543 ± 501	581 ± 868	476 ± 397
LNDFH I	1300	1460	750 ± 680			690 ± 290				777 ± 548
LNDFH II	170	230	860 ± 440			160 ± 120	120 (40–200)			
LDFT	330	480	70± 60	178 ± 184		450 ± 330	220 (0–340)			
LNT			3900 ± 1860	870 ± 623	880 ± 530	920 ± 650	1000 (770–2570)	1979 ± 738	1429 ± 693	1207 ± 599
LNnT			460 ± 360	149 ± 71	180 ± 85	1080 ± 1220	200 (0–1260)	1033 ± 445	765 ± 350	329 ± 153
LNFP I	2250	1640	280 ± 580	862 ± 734	910 ± 740	1910 ± 440	870 (0–1550)			1147 ± 802
LNFP II	230	290		359 ± 384		360 ± 190	200 (0–1260)			399 ± 377
LNFP III	340	370		248 ± 111		340 ± 410	330 (270–420)			
3′-SL	300	270	163 ± 105	165 ± 38	110 ± 35		200 (140–300)	100 ± 42	226 ± 107	154 ± 36
6′-SL	1770	1570	343 ± 235	558 ± 140	330 ± 140	470 ± 110	640 (530–970)	592 ± 219	621 ± 212	1297 ± 426
LST a	90	50	78 ± 60			90 ± 30	160 (120–230)	127 ± 86	104 ± 46	
LST b	60	70	84 ± 43				40 (20–150)			
LST c	60	310	620 ± 458			500 ± 100	370 (230–510)	941 ± 528	488 ± 224	367 ± 147
DSLNT	60	440	638 ± 484				320 (230–490)			644 ± 552

**Table nutrients-13-02272-t004c:** 

(c1) Mature	1 Month										
Reference Year	[43] 2010	[43] 2010	[34] 2015	[32] 2016	[2] 2017	[31] 2017	[30] 2017	[25] 2018	[24] 2019	[21] 2019	[21] 2019
Country	Germany (22 day)	Germany (1 month)	USA (28 day)	China (12–30 day)	Various (11–30 day)	Spain (16–30 day)	Singapore	China	China (28–34 day)	7 European countries (17 day)	7 European countries
2′-FL	3020	2960	1753 ± 1382	1900 ± 1200	2830 ± 500	2190 (0–3860)	2170 ± 832	1371 ± 1121	1399 ± 860	2627 ± 1028	2450 ± 935
3-FL	440	420	767 ± 654	570 ± 480	340 ± 90	1050 (0–1170)		894 ± 718	732 ± 545	594 ± 554	720 ± 608
LNDFH I	1550	1360			690 ± 200				626 ± 441	1275 ± 548	1105 ± 452
LNDFH II	260	240			140 ± 80	110 (60–250)					
LDFT	360	370	140 ± 165		370 ± 230	190 (0–350)				349 ± 379	277 ± 231
LNT			750 ± 481	620 ± 340	760 ± 410	1010 (770–2100)	979 ± 394	1225 ± 553	651 ± 316	1213 ± 720	1009 ± 591
LNnT			113 ± 71	120 ± 67	630 ± 850	180 (110–230)	263 ± 99	708 ± 299	237 ± 143	177 ± 97	153 ± 80
LNFP I	1720	1480	546 ± 512	540 ± 400	1370 ± 290	920 (0–1560)		1181 ± 578	701 ± 650	1431 ± 798	1071 ± 627
LNFP II	300	240	367 ± 350		320 ± 110	190 (10–1240)			275 ± 250	595 ± 630	549 ± 532
LNFP III	370	370	222 ± 77		320 ± 240	310 (190–460)				320 ± 141	311 ± 98
LNFP V				39 ± 41	80 ± 860				39 ± 41	124 ± 117	112 ± 99
3′-SL	260	270	146 ± 32	94 ± 25		180 (140–220)	217 ± 74	108.1 ± 37.4	651 ± 316	149 ± 38	141 ± 35
6′-SL	1420	1350	368 ± 108	250 ± 93	380 ± 90	650 (470–780)	561 ± 200	365 ± 160	736 ± 450	649 ± 189	465 ± 162
LST a	30	30			70 ± 50	180 (110–230)		58 ± 40			
LST b	90	100				50 (20–230)				80 ± 40	77 ± 38
LST c	250	240			240 ± 110	290 (190–440)		159 ± 111	173 ± 132	258 ± 128	148 ± 72
DSLNT	410	410				310 (220–510)			336 ± 222	385 ± 164	290 ± 135
6′-GOS/6′GL									22.3 ± 13.6	132 ± 47	26 ± 10
MFLNH-III/3-FLNH	200								416 ± 208	358 ± 192	
DFLNH/DFLNHa/DFLNH I/DFLNH II					2700 ± 3880			278 ± 163	227 ± 147		

**Table nutrients-13-02272-t004d:** 

(c2) Mature Milk	2 Months								3 Months			
Reference Year	[43] 2010	[32] 2016	[2] 2017	[30] 2017	[25] 2018	[25] 2018	[21] 2019	[26] 2018	[43] 2010	[2] 2017	[25] 2018	[21] 2019
Country	Germany	China (1–2 months)	Various (1–2 months)	Singapore	China	Malaysia	7 European countries	USA (42 day)	Germany	Various (2–3 months)	China	7 European countries
2′-FL	2820	1700 ± 1100	2390 ± 710	1764 ± 635	1176 ± 1019	1286 ± 1034	2075 ± 840	2480 ± 130	2590	2210 ± 710	984 ± 894	1819 ± 739
3-FL	560	720 ± 550	640 ± 150		1158 ± 864	762 ± 597	970 ± 692		670	670 ± 120	1366 ± 942	1140 ± 777
LNDFH I	1020		1100 ± 290				842 ± 327	1930 ± 50	1050	990 ± 290		719 ± 285
LNDFH II	190		190 ± 130						170	180 ± 130		
LDFT	380		380 ± 460				280 ± 155	240 ± 10	480	350 ± 230		273 ± 131
LNT		370 ± 220	1010 ± 530	633 ± 324	851 ± 319	1217 ± 651	700 ± 416	510 ± 30		730 ± 530	947 ± 602	599 ± 400
LNnT		83 ± 43	560 ± 1220	166 ± 72	569 ± 226	609 ± 285	128 ± 80			760 ± 1230	513 ± 419	108 ± 67
LNFP I	1060	340 ± 240	830 ± 450		950 ± 397	1660 ± 494	611 ± 423	580 ± 30	940	830 ± 440	1177 ± 679	469 ± 373
LNFP II	180		240 ± 190				474 ± 402		170	210 ± 150		433 ± 332
LNFP III	400		400 ± 410				358 ± 110		440	440 ± 410		353 ± 92
LNFP V		26 ± 25					91 ± 72					85 ± 66
3′-SL	230	80 ± 22		195 ± 60	99 ± 21	112 ± 28	129 ± 31	120 ± 0	240		114 ± 29	130 ± 35
6′-SL	630	140 ± 81	300 ± 150	280 ± 116	222 ± 105	251 ± 132	231 ± 101	250 ± 20	490	140 ± 130	137 ± 55	151 ± 87
LST a	10		10 ± 60		43 ± 32	60 ± 31			10	20 ± 50	40 ± 28	
LST b	80						64 ± 33		80			57 ± 31
LST c	110		130 ± 90		152 ± 118	130 ± 77	70 ± 47		90	110 ± 110	85 ± 67	44 ± 42
DSLNT	230						169 ± 83		210			136 ± 72
SLNFP I												
3′S3FL					8.5 ± 9.1	5.5 ± 3.5					7.5 ± 4.8	
LNH			80 ± 80					160 ± 10		120 ± 100		
MFLNH I/ 2 FLNH	130							110 ± 10	100			
MFLNH III/ 3 FLNH	120						208 ± 127		90			143 ± 94
DFLNH/DFLNHa/DFLNH I/DFLNH II			2840 ± 548				120 ± 97					98 ± 80

**Table nutrients-13-02272-t004e:** 

(c3) Mature Milk	4 Months					6 Months		
Study	[42] 2010	[32] 2016	[30] 2017	[27] 2018	[25] 2018	[25] 2018	[25] 2018	[23] 2019
Country	Samoa (22–155 day)	China (2–4 months)	Singapore	Canada (Caucasian and Asian mothers) (3–4 months)	China	China	Malaysia	UAE
2′-FL	690 ± 810	1300 ± 900	1376 ± 594	2256 ± 1846	866 ± 891	704 ± 752	1003 ± 803	997 ± 885
3-FL	2350 ± 1390	1100 ± 610		267 ± 171	1427 ± 892	1476 ± 790	1146 ± 869	1194 ± 106
LNT	1310 ± 590	290 ± 170	407 ± 200	1047 ± 479	866 ± 443	785 ± 497	867 ± 426	504 ± 337
LNnT	200 ± 290	65 ± 39	108 ± 76	285 ± 246	525 ± 315	446 ± 234	571 ± 321	250 ± 188
LNFP I	350 ± 450	180 ± 140		788 ± 754	1228 ± 557	945 ± 436	1036 ± 492	650 ± 416
LNFP II	2770 ± 2140			1853 ± 879				
LNFP III				92 ± 51				
LNFP V		25 ± 25						
3′-SL	133 ± 56	79 ± 20	198 ± 59	361 ± 231	126 ± 36	127 ± 39	135 ± 51	134 ± 69
6′-SL	189 ± 265	78 ± 40	120 ± 45	162 ±128	97 ± 33.2	83 ± 54	84 ± 34	91 ± 108
LST a	44 ± 62				36 ± 23	33 ± 17	84 ± 55	11 ± 8
LST b	193 ± 215			118 ± 69				
LST c	201 ± 316			43 ± 42	56 ± 41	47 ± 58	145 ± 160	
DSLNT	317 ± 409			315 ± 246				
3′S3FL					9.1 ± 7.5	10.5 ± 6.9	9.0 ± 6.5	10 ± 14
6′SLN					4.0 ± 2.3	2.0 ± 2.1	3.6 ± 2.8	5 ± 1

^1^ Sum of LNFP II and LNFP III. ^2^ Sum of all LNFP. nr, not reported; 2′- FL, 2′-fucosyllactose; 3-FL, 3-fucosyllactose; DFLac/LDFT, Lactodifucotetraose; DFLNT, Difucosyllacto-*N*-tetrose; DFLNH I, Difucosyllacto-*N*-hexaose I; DFLNH II, Difucosyllacto-*N*-hexaose II; LNDFH I, Lacto-*N*-difucohexaose I; LNDFH II, Lacto-*N*-difucohexaose II; LNnDFH, Lacto-*N*-neodifucohexaose; LNFP I, Lacto-*N*-fucopentaose I; LNFP II, Lacto-*N*-fucopentaose II; LNFP III/LNnFP II, Lacto-*N*-fucopentaose III/ Lacto-*N*-fuconeopentaose II; LNFP V, Lacto-*N*-fuconeopentaose V; LNnFP I, Lacto-*N*-neofucopentaose I; LNnFP V, Lacto-*N*-neofucopentaose V; 3′-GOS/3′-GL, 3′-galactosyllactose; 6′-GOS/6′-GL, 6′-galactosyllactose; LNT, Lacto-*N*-tetraose; LNnT, Lacto-*N*-neotetraose; LNH, Lacto-*N*-hexaose; LNnH, Lacto-*N*-neohexaose; MFLNH I/2-FLNH, Monofucosyllacto-*N*-hexaose I; MFLNH III/3-FLNH, Monofucosyllacto-*N*-hexaose III; 3′SL, α2,3-Sialyllactose; 6′SL, α2,6-Sialyllactose; 3′S3FL, 3′-sialyl-3-fucosyllactose; 6′SLN, 6′-sialyl lactosamine; LST a, α2,3-Sialyllacto-*N*-tetraose a; LST b, α2,6-Sialyllacto-*N*-tetraose b; LST c, α2,6-Sialyllacto-*N*-tetraose; DSLNT, Disialyllacto-*N*-tetraose; S-LNnH II; α2,6-Sialyllacto-*N*-neohexaose, A-Tetra, A-tetrasaccharide; SLNFP I, Sialyllacto-*N*-fucopentaose I; SLNFP II, Sialyllacto-*N*-fucopentaose II.

**Table 5 nutrients-13-02272-t005:** Summary of changes in individual HMO concentrations throughout lactation.

Increasing over Lactation	Decreasing over Lactation	No Significant Trend
3-FL (3-Fucosyllactose) 3′S3FL	2′-FL (2′-fucosyllactose) LNT (Lacto-*N*-tetraose) LNnT (Lacto-*N*-neotetraose) LNFP I (Lacto-*N*-fucopentaose I) Lacto-*N*-fucopentaose III (LNFP III) Lacto-*N*-fucopentaose V (LNFP V) 3′-SL (α2,3-Sialyllactose) ^1^ 6′-SL (α2,6-Sialyllactose) LST a/b (Sialyllacto-*N*-tetraose) LST c (Sialyllacto-*N*-neotetraose)	LDFT (Lactodifucotetraose) LNFP II (Lacto-*N*-fucopentaose II) A-Tetra LNnFP II (Lacto-*N*-neofucopentaose II) DSLNT

^1^ Results from Ma et al., (2018) [25] study and Huang et al., (2019) [24] study showed 3′-SL increased throughout lactation in Malaysia and Chinese human milk.

**Table 6 nutrients-13-02272-t006:** Percentage of secretors in different countries.

Country	Total Number	Secretors (%)	Reference
USA-Washington state	41	68	[6]
USA-California (Hispanic)	19	95	[6]
USA	79	68	[8]
	45	66–77	[29]
Canada (73% are Caucasian)	427	72	[27]
Chile	44	84	[8]
Mexico	156	100	[8]
Peru	43	98	[6]
Germany	30	73	[43]
	18	83	[8]
France	22	91	[8]
Italy	29	86	[8]
Spain	41	76	[6]
Sweden	7	100	[8]
	24	79	[6]
China	20	67	[25]
	30	73	[1]
	32	78	[8]
	650	79	[32]
Philippines	22	46	[8]
Singapore	26	72	[8]
Republic of Malawi	88	78	[29]
Ethiopia rural	40	65	[6]
Ethiopia urban	40	78	[6]
Kenya	42	81	[6]
Gambia rural	40	65	[6]
Gambia urban	40	85	[6]
Ghana	40	68	[6]
United Arab Emirates	81	74	[23]

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
