# Peer review of "Changes in HMO Concentrations throughout Lactation: Influencing Factors, Health Effects and Opportunities"

_nutrients, 2021, doi:10.3390/nu13072272_

Round 1

Reviewer 1 Report

This work deals with mother's milk oligosaccharides composition, by compliling several previous studys and reviews. As metioned by authors, the task is particulrly hard owing to different analytacal methods used, type of HMOs inestigated, period of lactation. Langage is fair. A few mistakes/ phrasing should be fixed.

Please consider these main issues :

  • Authors should be more explicit regarding their criterions/methodology (based on impact factor or other kind of « visibilaty/reliability » indicator...) to select previous articles on the topic. It could be interesting to draw a rapid state of the art about knowledge of mother's milk carbohydrate composition. Why no article published before 1999 was taken into account ?

  • Authors should compare the different methods used by authors cited in their work, to evidence possibly experimental bias. A simple Table that relates advantages and drawbacks of each method may be useful. All in all, which method authors would recommend so as to standardize lab practices ?

  • It could be suggested to take as illustrative examples some typical changes in mother's milk compositions according to lactation period, while focusing on the 2, 3 main HMOs (otherwise the 3 main classes of HMOs : Fucosylated, Sialylated, neutral) accounting for 90% (at least) of total composition, so as to evidence worldwide differences in milk from women living in e. g. three countries located in different continents.

  • With regards to typical compositions of mother's milk according to geographical area/lactation period, which optimal composition of formulated (i. e. commercial) mother's milk should be recommended by authors ? This may be a good sum-up.

  • Some SD values are peculiarly wide owing to natural variation of the data of interest (HMO concentrations) and sampling size, differing from one study to one another. It vould be suggested to the authors to display more preferably intervals of confidence, if possible.

  • Be careful with citation numbering. There is apparently a bug with references : « Error! Reference source not found ». Please check this.

Specific comments are listed below. Please read them carefully to improve the clarity and reliability of the manuscript.

Hoping that you will agree with most of these suggestions,

One reviewer

Line 26 : could you specify a range of concentrations for both types of compounds ? Otherwise a relative amount could be informative (e. g. x-fold higher than...)

28 : recognized ? American or british english required by the journal ?

30 : maybe « monosaccharide » (or carbohydrate monomers) is better than « sugar ».

34, 59, 60 : characterized ?

35 : could you provide a study wherein the factors (environmental, mother's food, genetics...) influencing the HMO composition are investigated.

37 : synthesizes ?

38 ok well written.

49 summarizes?

67 : why did you choose specifically this previous review ? Split into 2 sentences : « … 2017[2]. Additionally, data from 12 (peer-reviewed?) articles published between 2016-19 provided...

Which criterions did you use to select these previous data, e. g. impact factor ? Did these research articles published in several representative journals within the infant nutrition field ? How many different research teams (or labs in the world) were involved ? This may help the reader to figure out the number of independant sources used !

74-75 : good remark, also to evidence the possibility of a conflict of interest. Have you got an idea of the total number of relevant studies, to the best of your knowledge (or within your library), regarding this topic ?

Table 1 : what « N » means ? Number of milk samples ?

By comparison, have you found an elder study than those published in 1999 ?

Interestingly, data seems to be scarce coming from Africa, except South Africa, Central Asia, India and South America, whereas these parts of the world were affected by children undernourishment. Have you got an explanation ?

101 : do you know the differences regarding breastfeeding period between the mentioned countries ?

105 : another explanation may arise from the number of arget HMO, provided these selected covered a great majority of those naturally found in breast milk. Is it possilbe to show in Table 1 the number of HMO (or a percentage of total) investigated in each study ?

112 : could you inform the reader about the differences (some figures please) in times of milk collection ?

123 : analyzed

131 : okay, however it may be more interesting to compare the relative amounts of HMO studied (as % of total) rather than their absolute number.

135 : disparities

139 : add a slash « and/or »

Table 3 : add « mature » in the first row. Please change the size, since the right side is cut by the edge of the page. Why data is not available (« nr ») from the studies of Azad (18) and Xu (29) ? Aren't they relevant ?

145 : « summarizes »

155 : what should be the « best » method ? Otherwise you can select the three mains of them and compare advantages/drawbacks, so as to give a direction to further researches – non-standardization of the methods is an issue to conduct accurately comparative studies.

159 : « lower than either LNT or LNFP 1 » or lower than the summed concentrations of both» ? Please specify.

175 : which substrate ? Where does it come from ?

179 : you wrote « for substrate ». Do you mean one specifically ? Which one ?

180-182 : is there any hypothesis to explain such limitation ?

192 : what is « FUT 2 » ? I did not find it in Table 1

198-200 : give some figures please, or you could write « The conc. of compound C decreased X-fold rom the early stages of lactation to... »

204 : apparently a langage mistake : write « have an impact on » or « impact the proportions ».

208-209 : some figures please !

So as to illustrate your purpose, is it possible to dra a chart displaying the changes in concentration of e. g. the three mains HMO (or the sum of those representing 90% of total HMO) as reported by Thurl et al. ?

Table 4 : is day 3 concerned only references 21 ; 44, 45 ; 43, or 25 ; 21 ; 44, 45 ; 43 and 26 ? please change a bit the presentation in the 2 first rows. Why weren't the SD values displayed for ref 43 ?

Please specifiy the references wherein the secretor status is known or not.

204-215 : I strongly suggest you to add some figures for the reader to figure out an order of magnitude.

Figure 1 : did you draw the charts from reported values previously or you copied and pasted such presentation ? Please specify. Could you add SD values ?

Table 4 :Regarding summed lines LNFP II + III, I am afraide there is mistakes regarding SD values, larger than the average : 380 +/- 2401 (!) ; 422 +/-518 ; 260 +/-290. Please check.

Is there any explanation regarding the large SD values ? The more values are inculded in the mean, the larger are SD values. Maybe you should calcultate a confidence interval for each population. Do you think it will be more meaningful ?

Table 5 : we understand your attempt to synthesize this huge set of values, however you should draw a (scatter) chart with the « most representative (average) values » (let's say based on the study of Thurl et al. for eg 2-3 countries located in different continents) so as to ensure better clarity of the purpose ? Did you notice differences of magnitude (not the tendencies, as you seemed to indicate) regarding these trends according to the country ?

232 : For my point, Figure 2 (A) should be deleted, (B) is more informative regarrding changes in overall composition. The sentence « Percentage of fucosylation increased while percentage of sialylation decreased and non-fucosylated neutrals remained stable over time. » is a comment and should added to the text (not in the caption).

236 : « Secretor and Lewis blood group status » : did you refer to one Table or Figure ? Please specify.

237 : « from among the more » sounds weird. Please rephrase.

239-242 : maybe you should define the secretor status in the Intro, since this was mentioned in Table 2.

274-275 : given thaht the SD between 6.5 ± 1.7 g/L and 5.2 ± 2.5 g/L crossed, are you sure mean values are significantly different (i.e. sufficiently spread) ? I am afraid intervals around mean values are within the experimental error, as it could be evidenced by an ANOVA. Please check.

276 : same remark for 4.9 ± 1.2 and 3.4 ± 2.3 g/L ; these appeared not to be significantly different given that large SD values, regardless of the stat test used (Newman and Keuls, Duncan...)

291 : « larger » than ?

293 : maybe I asked you before, but what is the source of this substrate ? Is it related to mother's food intake ?

page 20

From here, Please add line numbers !

Section 3.2. « Country of origin »

- « the data presented from each study « . Write « presented (or displayed) in each study »

  • « different subpopulations it is «  add a coma before « it »

  • « standardised and validated interlaboratory methods « . Standardized ? This passage is well-written and a global issue regarding the intrinsic variability in life sciences. Different analytical methods add experimental errors. Could you recommend one peculiar method ?

p. 21

  • first line « analyzed ». Add « milks » after « women's ».

  • Add a coma : « … and LNFP III, while... »

  • « higher levels » than ? Those in finnish and chinese women's milks ?

  • « … lower abundance... » than ? in finnish and south african women's milks ?

  • « [28] and the distribution of secretors... ». Write « [28], likewise the distribution of... ».

  • « and in the USA ».

  • Write « … the USA ( at 2210 – 4130 mg/L, 2061 – 3370 mg/L and 1753 – 3020 mg/L in colostrum [21,26,31,34], transitional [31,34,43,64] 1-month mature milk, respectively)

  • add « to those in samples »

  • « … from women in Asian regions, i.e. China, Malaysia, Japan, Singapore and Samoa (at 1580 – 2490 mg/L, 220 – 2000 mg/L,and 1371 – 2170 mg/L in colostrum [24,25,44,45], transitional [24, 25, 32, 42] and in 1-month mature milk [24,25,30,32], respectively.

      1. Maternal physiological status

- What's the meaning of the abbreviation « BMI » ? Body mass index ?

- « Substrate availability » - which one ?

- Did authors evidenced a possible link between defiency in one essential amino acid in mother's food intake and enzyme expression involved during milk biosynthesis ?

- Move above « body mass index ».

- Rephrase : « … by enzyme expression and activity and substrate availability, whereas maternal factors may influence maternal physiology and glycosylation within the initial period of lactation

p. 22

  • Is there no data regarding lacking HMO in undernourrished women ? All in all, which compounds are reported to be essential in mother's milk according to infant age ? If not, I am a bit surprised that such a field was not explored previously. Maybe I suggest you to have a look on studies focusing on oligosaccharides composition of several mother's milks available in the current market, so as to evidence on which basis formulations were established – another question may be whether commercial mother's milk were different from one continent to one another.

  1. « Health effects of HMO »

    - «  indigestible characteristics » this means that HMOs were prebiotic substances ?

    - « good » bacteria : which ones ? Please specify

    1. « HMO profile and microbiota in infant’s gut and mother’s milk »

      - microbiota in milk : do you mean bacteria already present in mother's milk ?

      - add « … infants fed with non secretor milk... »

      - « … the colonisation by these beneficial microorganisms and more Clostridium and Enterobacteria in their faeces » is it detrimental for infant's health ?

p. 23

  • What does « MING » mean ?

  • «  higher bacterial counts » than ?

  • « … oligosaccharides (LNDFH, LNH, LNFP I) » which relative amount of total HMOs these represented ?

  • What is the «microbiota diversity » ?

  • « However overall » sounds weird. Please rephrase.

  • Maybe « correlation » is better than « association ».

  • Rephrase : « … the overall milk fatty acid profile, including in particular individual long chain fatty acids [22:6n3, 22:5n3, 20:5n3, 17:0, 18:0] and milk microbiota composition.

  • « standardized »

  • Good conclusion for this section. Your paper may help to solve methodology discrepancies among authors, make a few recommendations  to go further the debate!

    1. « Supplementation of infant formula with HMOs »

      - Please rephrase : « … higher than those found in ruminant milk ».

      - Rephrase : « … lack of oligosaccharides in terms of quantity and diversity »

p. 24

  • which physiological parameters regarding infants were considered in ref 18 to evaluate the « growth of infants (consuming formula containing 2'-FL) » ?

  • « both doses » : please calculate the level of supplementation in 2'-FL as compared to control. Could you compare such supplementation with experimental values found in natural mother's milk as listed in Table 1, in the same (or close) geographical area ?

  • « supplemented with 2 HMOs, 2’-FL and LNnT » : which concentrations ? Could you compare with average natural levels ?

  • Add « … compared to infants fed with the formula containing no HMOs... »

  • Change « … The study showed a correlation between feeding the 2-HMO supplemented.

  • Please rephrase « and towards those »

  1. Concluson

p. 25

  • « parity » what do you mean ?

  • « followed by slightly lower concentrations in transitional milk, and a gradual decline in mature milk as lactation progresses ». You may add that the changes in HMO composition on infant's health should be considered with respect to weaning age, depending on geographical area

  • Rephrase « … only present in only minimal amounts »

  • Add « Infants fed with non-secretor milk ».

  • Splt into two sentences « by these beneficial microorganisms . In contrast, these infants may show an increasing resistance... »

  • « Supplementation of infant formula « : which levels do you recommend to formulate commercial mother's milk ?

Author Response

Thank you for your time and comments.

Point-by-point comments attached.

Cheers

Caroline

Reviewer 2 Report

The manuscript presents a thorough review of the literature concerning HMO concentrations in breast milk and how they change throughout the course of lactation. Additionally, it includes a more limited review of the association of individual subsets of HMOs with predictors such as maternal diet and health and with outcomes such as infant intestinal microbiota. Overall, the manuscript is well-written and informative. My only significant concern is that the authors give only cursory mention to the role of commercial interests in existing scientific research regarding HMOs, and seem to imply that a preponderance of commercially-funded research on a topic is somehow beneficial. In fact, the preponderance of commercially-funded research regarding HMOs, especially with respect to supplementation of infant formula, has the potential to introduce significant bias to this area of science, and this should be identified clearly in any review of the topic. Other smaller concerns are listed below.

Major concern:

  • As mentioned above, in page 2 lines 72-75, there is an implication that the preponderance of commercial funding is beneficial. The authors are to be commended for noting the preponderance of commercial funding in the manuscript. However, this should be accompanied by a discussion of the limitations introduced by a preponderance of commercial funding.

Minor comments:

  • Lines 31-33 appear to be either a sentence fragment or with incorrect punctuation, so it is difficult to understand their meaning.
  • Overall, my version of this manuscript contains the message “Error! Reference source not found” in multiple locations, often with an accompanying sentence fragment. I am therefore unable to understand some of the text and cannot comment on it.
  • Table 3 is truncated in the version of the manuscript I received, so I am unable to comment on the left-hand columns which I cannot see.
  • For Table 4, a justification should be given for including only data from secretor mothers or mothers with unknown secretory status (i.e., excluding mothers known to be nonsecretors), or alternatively data from nonsecretor mothers could be included.
  • Some of the results in Table 4 overlap two or more rows (e.g. 3563 ±1920, 750 (0-1190), etc.) I am unsure whether this is intentional or erroneous. If erroneous, it should be corrected. If intentional, a footnote indicating the reason for it would be helpful.
  • For Figure 2, it is unclear why the authors include only data previously compile by Thurl et al., which was published in 2017, and do not include subsequent data described in the review.

Author Response

(The authors gave the same response as above.)

Round 2

Reviewer 1 Report

see attachment

Author Response

Thank you for your time to review this manuscript.